# Palmitoylethanolamide: A Natural Compound for Health Management

**DOI:** 10.3390/ijms22105305

**Published:** 2021-05-18

**Authors:** Paul Clayton, Mariko Hill, Nathasha Bogoda, Silma Subah, Ruchitha Venkatesh

**Affiliations:** 1Institute of Food, Brain and Behaviour, Beaver House, 23-28 Hythe Bridge Street, Oxford OX1 2EP, UK; 2Gencor Pacific Limited, Discovery Bay, Lantau Island, New Territories, Hong Kong, China; mariko@gencorpacific.com (M.H.); nathasha@gencorpacific.com (N.B.); silmah@gencorpacific.com (S.S.); 3School of Medicine, University of Hong Kong, Hong Kong, China; ruchi.v@hotmail.com

**Keywords:** inflammation, neuroinflammation, immunomodulator, pain, ALIAmides, endocannbinoids, bioactive lipids, immunomodulation, molecular mechanism, nutraceuticals, dietotherapy, lipid signaling

## Abstract

All nations which have undergone a nutrition transition have experienced increased frequency and falling latency of chronic degenerative diseases, which are largely driven by chronic inflammatory stress. Dietary supplementation is a valid strategy to reduce the risk and severity of such disorders. Palmitoylethanolamide (PEA) is an endocannabinoid-like lipid mediator with extensively documented anti-inflammatory, analgesic, antimicrobial, immunomodulatory and neuroprotective effects. It is well tolerated and devoid of side effects in animals and humans. PEA’s actions on multiple molecular targets while modulating multiple inflammatory mediators provide therapeutic benefits in many applications, including immunity, brain health, allergy, pain modulation, joint health, sleep and recovery. PEA’s poor oral bioavailability, a major obstacle in early research, has been overcome by advanced delivery systems now licensed as food supplements. This review summarizes the functionality of PEA, supporting its use as an important dietary supplement for lifestyle management.

## 1. Introduction

The nutrition transition has created multiple pandemics of chronic degenerative disease, which are overwhelming health care systems everywhere [1,2,3,4,5]. Deleterious lifestyle changes include exposure to environmental and industrial pollutants, the erosion of photoperiodism, decreased physical activity, high intakes of ultra-processed foods and a resulting increased dependence on pharmaceuticals [1,2,3,4]. These create dysnutrition, chronic inflammation, dysbiosis and immunosenescence, which in turn increase susceptibility to metabolic disease, cardiovascular disease, cancers, chronic pain, joint disease, asthma, gastrointestinal disorders, affective disorders and neurodegenerative disease [1,2,4,5,6,7].

Many pharmaceutical products have been developed to treat the symptoms of these disorders, but they do not address their fundamental causes and are too toxic to be used preventatively [5,8]. Foods and food derivatives are eminently suitable for public health interventions.

Due to their generally wide therapeutic indices, there is a clear and growing rationale to use supplementation to recreate pre-transitional dietary profiles, restore nutritional and metabolic normality [9,10,11,12,13] and salvage our public health. Such supplements should ideally protect against inflammatory and oxidative stress, and in the current public health environment, they should also target pathways involved in pain sensation, immune regulation, recovery and brain health.

Palmitoylethanolamide (PEA) is an endocannabinoid (eCB)-like bioactive lipid mediator [14,15] belonging to the N-acyl-ethanolamine (NAE) fatty acid amide family [16]. Synthesized on demand within the lipid bilayer [17], it acts locally [18] and is found in all tissues including the brain [18,19]. PEA is thought to be produced as a pro-homeostatic protective response to cellular injury [18] and is usually up-regulated in disease states [18,20]. Its pleiotropic effects include anti-inflammatory, analgesic, anticonvulsant, antimicrobial, antipyretic, antiepileptic, immunomodulatory and neuroprotective activities [14,17,21,22,23]. PEA’s multi-faceted effects are due to its unique mechanisms of action that affect multiple pathways at different sites [18]; primarily targeting the nuclear receptor peroxisome proliferator-activated alpha (PPAR-*α*), PEA also acts on novel cannabinoid receptor, G protein-coupled receptor 55 (GPR55) and G protein-coupled receptor 119 (GPR119) [9,14]. Moreover, it indirectly activates cannabinoid receptors 1 and 2 (CB1 and CB2) through inhibiting the degradation of the endocannabinoid, anandamide (AEA), a phenomenon known as the ‘entourage effect’ [9,16].

Additionally, PEA activates and desensitizes the transient receptor potential vanilloid receptor 1 (TRPV1) channels, contributing to a significant anti-nociceptive effect. It does so via several mechanisms, including the entourage effect, through PPAR-*α* activation and by potentially acting as an allosteric modulator [9,16]. PEA’s inhibition of mast cell (MC) activation also plays a role here, a mechanism discovered by Professor Rita Levi Montalcini and colleagues, who characterized this as Autacoid Local Inflammation Antagonism (ALIA) [23,24,25]. Detailed information into PEA’s mechanisms of action can be found elsewhere [9,23,24].

PEA’s multiple mechanisms of action generate therapeutic benefits in many disorders, including allergic reactions [17,21,25,26], influenza [24], common cold [24], chronic pain [27,28], joint pain [29], psychopathologies [27,30] and neurodegeneration [14,15,16,17,19]. They also contribute to enhanced muscle recovery [31] and improved cognition [26,32], mood [26,28,33] and sleep [34,35]. Endogenous PEA is generally insufficient to counter chronic allostatic load as seen in chronic inflammatory disorders [30,36,37], making exogenous administration a viable therapeutic strategy to top-up endogenous levels and restore bodily homeostasis [36,37].

A number of PEA-containing products are licensed for use as nutraceuticals, food supplements or foods for medical purposes in different countries, with a generally recommended dose of 1200 mg/day [16]. A novel form of PEA using LipiSperse^®^ technology to achieve enhanced bioavailability will likely allow lower dosages (See Section 9). These products testify to PEA’s excellent clinical and pre-clinical safety record, which goes back to the 1950s [18,24,25].

Table 1 lists the applications of PEA in physical and mental health management.

## 2. PEA and Immunity

Emergent viruses underline the importance of immune health. In the absence of vaccines, the body’s innate immune system is the primary defense against pathogens [86]. The immune system and especially the innate components are also key to self/non-self differentiation and recognition, and to proteostasis and tissue repair [87].

Innate cellular components include phagocytes (e.g., macrophages), neutrophils, MC and others [87]. These and innate humoral components, including cytokines, modulate the inflammatory response, maintaining immune-surveillance and the acute inflammatory reaction. Conversely, the modern diet and the ageing process create imbalances which drive chronic inflammation [88]. The synthetic anti-inflammatory drugs have applications in the short-term management of chronic inflammation but are too toxic for long-term and/or preventive use at the public health level. Endogenous and/or food-derived supplements offer a safer alternative [89].

In the 1960s, PEA was first marketed for prophylactic treatment of influenza and the common cold. Research interest increased in the 1970s, with six clinical trials confirming the effectiveness of PEA on influenza symptoms and incidence. Pre-clinical work had suggested that this prophylactic action was due to PEA’s ability to increase the body’s nonspecific (innate) resistance to bacteria and viruses [90]. The immunomodulatory effects of PEA are summarized in Table 2.

### 2.1. PEA’s Antiviral Activities

PEA binds to the PPAR-*α* receptor and initiates mechanisms which activate macrophages and, thus, enhance resistance to infection. Influenza infection is characterized by increased production of inflammatory cytokines such as tumor necrosis factor *α* (TNF)-*α*, interleukin- (IL-) 1, IL-6, and IL-10, which may lead to a hyper-inflammatory state known as a ‘cytokine storm’ [94]. This contributes to the increased morbidity and mortality associated with virulent infections [95]. PEA’s binding to PPAR-*α* receptors of immune cells such as macrophages and MCs leads to reduced production of inflammatory signals and reduced pain signals [38], as documented in over 60 PubMed indexed papers. PEA also modulates interleukin chemistry. Di Paola et al. demonstrated that 10 mg/kg body weight PEA significantly reduced intestinal damage and inflammation in a reperfusion injury murine model, inhibiting proinflammatory cytokine production (TNF-*α*, IL-1*β*), adhesion molecule (ICAM-1, P-selectin) expression and NF-*κ*B expression [39]. PEA’s anti-inflammatory and cytokine modulating actions explain its documented ability to provide symptomatic relief at the onset of influenza and common cold.

PEA also modulates MC activity. MCs are strategically localized at sites that directly interface with the external environment and act as sensors/transponders, detecting different kinds of injuries and responding with varying degrees of activation. When activated, they regulate innate and adaptive immune responses by releasing pre-formed and newly synthesized mediators [96]. MC hyper-activation, which can be disabling and even lethal, is treated with anti-histamines and anti-leukotrienes. These drugs have serious side effects, including vomiting, heartburn and immune suppression [97,98].

Cannabinoid-based drugs have shown promise but are beset by ethical problems [96]. PEA prevents MC degradation in vitro and in vivo, and down-regulates MC-derived inflammation [40]. More recently, PEA was shown to inhibit release of histamine, prostaglandin D_2_ and TNF-*α* induced by allergens in isolated canine skin MC in a dose-dependent manner ranging from 10-8 to 10-6 M of PEA concentration [41]. In this way, PEA can be used as an immune checkpoint.

### 2.2. PEA’s Antibacterial Activities

PEA protects against bacterial infection via innate immune modulation involving MCs, macrophages and microglia. Prophylactic PEA at a dose of 0.1 mg/kg body weight prolonged survival rate and reduced neuro-inflammation in an aged murine bacterial meningitis model, in the absence of antibiotics [42]. In the early phase of infection, the PEA pre-treated mice showed lower bacterial titers in spleen, liver and blood than controls. PEA pre-treatment also increased the survival rate and bacterial clearance of immunocompetent young mice challenged with *E. coli* [42]. Other pre-clinical studies have similarly shown PEA’s ability to increase resistance to systemic bacterial infections [43,44].

Microglia cells, the resident macrophage population in the central nervous system (CNS), defend the CNS from infection [99], clear cellular debris [100] and provide various support functions to neurons. Exposure to PEA stimulates E. coli uptake in macrophages and microglia [91], likely via CB2-enhanced chemotaxis and PPAR-*α-*induced phagocytosis [92]. PEA’s ‘entourage effect’ enhances the physiological effects of endocannabinoids such as AEA by preventing their enzymatic-mediated hydrolysis by fatty acid amide hydrolase (FAAH), which results in TRPV1 and CB2 stimulation. This provides another route to the activation of macrophages, neutrophils and other immune cells [93], thus contributing to PEA’s anti-infective properties. AEA also has anti-inflammatory and pro-apoptotic activities through which it can inhibit TNF-*α* and NF-*κ*B [101]. PEA is considered to constitute a “parallel” endocannabinoid signaling system without the adverse effects of synthetic endocannabinoids.

### 2.3. PEA’s Modulation of Gut-Derived Immunity

The majority of immune cells are localized within the gut-associated lymphoid tissue (GALT) [87]. When food is ingested, the body is exposed to abundant antigenic stimulation, requiring the immune system to discriminate between potential pathogens, and food proteins and symbionts. The gut microbiota plays an important role in nutrient metabolism and absorption. It is a critical factor in determining gut health, providing energy for epithelial cells, regulating local and systemic immune function and maintaining epithelial barrier integrity [102]. Acute and chronic gastrointestinal tract (GIT) inflammation caused by dysbiosis or vitamin D deficiency damages the epithelial barrier, known colloquially as ‘leaky gut’. This triggers efflux of immune cells from GALT, causing immune dysregulation and a range of pathologies [103].

PEA may contribute to correcting the effects of dysbiosis. In an induced inflammation state, such as vitamin D deficiency in mice, intraperitoneal administration of PEA increases the level of commensal bacteria such as *Akkermansia muciniphila* (protective effects against obesity and diabetes), *Eubacterium* (microbiome regulatory properties) and *Enterobacteriaceae* [83]. Moreover, exogenous administration of PEA relieves chronic and acute GIT inflammation via its action on PPAR-*α* in the colon [84]. For these reasons, PEA is becoming an increasingly popular topic in microbiome research.

### 2.4. Future Directions of PEA’s Role as An Immunomodulator

The beneficial effects of PEA on immune function are documented in over 350 peer-reviewed papers, including animal and human studies [38]. These show that PEA’s multifaceted immunomodulation reflects its ability to target multiple pathways which work synergistically and physiologically to produce therapeutic effects [104].

PEA’s safety/efficacy profile also makes it suitable for prophylactic use. Unlike other endocannabinoids, catabolism of PEA in the body gives rise to relatively inactive products (palmitic acid and ethanolamine), which do not produce adverse effects [105]. Regular administration of PEA can, therefore, be used to maintain immune health in programs designed to support health in general and healthy ageing.

## 3. PEA and Inflammatory Reactions

### PEA’s Antiallergic Effects

Allergic reactions such as allergic rhinitis, allergic dermatitis and allergic asthma are characterized by inflammation and inflammatory cell infiltration [106,107,108]. Upon recognition of an allergenic factor, MCs are activated and degranulate, releasing mediators such as histamine, cytokines and chemokines, which trigger an inflammatory response. The antiallergic effects of PEA can be traced back to the 1950s, when Coburn and colleagues reported that a phospholipid fraction isolated from egg yolk demonstrated antiallergic activity in guinea pigs [45].

More recent animal studies have confirmed PEA’s antiallergic actions, which include down-regulation of MC recruitment and degranulation. PEA’s protective effects are mediated by its cellular targets, including the direct activation of PPAR-*α* and GPR55 receptors and the indirect activation of cannabinoid receptors (CB1 and CB2) and TRPV1 channels [46]. In one study conducted on canine skin mast cells, PEA (in doses ranging from 10-8 M to 10-5 M) induced a significant and dose-dependent inhibition of PGD_2_, TNF-α and histamine release [41]. In Ascaris hypersensitive beagle dogs, a single oral dose of um-PEA (at doses of 3, 10 and 30 mg/kg) significantly reduced allergic wheal reactions in skin [47]. Treatment with PEA also showed improvement of clinical signs in cats with eosinophilic granuloma [48]. Another study showed that treatment with PEA was effective in the improvement of skin lesions and pruritus in dogs with atopic dermatitis and moderate pruritus [49]. In mice sensitized with aerosolized ovalbumin, bronchial levels of PEA were reduced, while CB2 and GPR55 were up-regulated [46]. Leukocyte infiltration and pulmonary inflammation were significantly inhibited by 10 mg/kg PEA supplementation prior to sensitization. Furthermore, pulmonary mast cell recruitment and degranulation, and leukotriene C_4_ production were also significantly inhibited, demonstrating a depletion/repletion scenario.

Mice with 2,4-dinitrofluorobenzene (DNFB)-induced contact allergic dermatitis (CAD) showed increased ear skin PEA levels, and increased TRPV1, PPAR-*α* and NAPE-PLD in ear keratinocytes in the early stage of the condition [50]. In this model, PEA given intraperitoneally exerted anti-inflammatory effects, likely through TRPV1 receptors. The authors also report findings from in vitro experiments that exogenous PEA treatment inhibited the expression and release of the inflammatory cytokine MCP-2 in poly-(I:C)-treated HaCaT cells, again likely through a TRPV1 mediated mechanism [50]. In late-stage CAD, 5 mg/kg PEA administration reduced numbers of MCs in the ear while inhibiting VEGF and its receptor Flk-1, components in angiogenesis, which is a major characteristic of contact hypersensitivity reactions [51]. Vaia and colleagues also reported that PEA administration led to the restoration of levels of 2-AG. This, together with findings of the reduction in MC activation and neo-angiogenesis being blocked by a CB2 receptor antagonist, points to a CB2 receptor mediated mechanism of anti-inflammatory action [51]. Abramo et al. reported increased expression of CB1 and CB2 receptors in the lesional skin of dogs with atopic dermatitis compared to normal dogs [52]. These findings highlight the importance of PEA as an anti-inflammatory and protective modulator.

There are a few studies demonstrating the efficacy of PEA formulations in allergic conditions in humans. Findings from a multinational, multicenter study on atopic eczema showed that topical application of a cream with a unique lamellar matrix containing PEA significantly reduced intensities of erythema, pruritus, excoriation, scaling, lichenification and dryness [53]. There is also an intriguing case report of a 13-year-old child with autism, presenting with significant atopic illness including chronic eczema, allergic rhinitis and asthma. The physicians reported that daily oral administration of PEA for one month (first at 600mg/day and later increased to 1200mg/day) resulted in a marked reduction in allergy stigmata, skin eczema and urticaria [26]. There is a need for further clinical studies to confirm these findings.

## 4. PEA and Brain Health

Neuroinflammation is now regarded as a key aspect in the pathogenesis of many if not all neurodegenerative disorders [15,18] including Alzheimer’s (AD) [14,16,17,18,19,54,55], Parkinson’s disease (PD) [32,109], stroke [56], traumatic brain injury (TBI) [20,57,58,81], autism spectrum disorder (ASD) [26,59], epilepsy [21] and cognitive [32], behavioral [26,59] and mood disorders [28].

Key inflammatory cells in these conditions include MCs, microglia and astrocytes [55]. While the detailed biochemistry of neuroinflammation is beyond the scope of this review, the neuroinflammatory cascade degrades neuronal structures, impairing neuronal function and viability [18,54]. Dampening neuroinflammation may, therefore, be an effective way to preserve brain structure and function [55,109]. There is evidence that NAEs such as PEA exert neuroprotective actions by promoting the resolution of neuroinflammation [33]. The increased PEA levels in several CNS pathologies may, therefore, be an endogenous cytoprotective response to injury [18,21,28]. PEA and its receptors are present in different cell types in the CNS [17,18], and as PEA crosses the blood brain barrier [110], additional exogenous PEA may find uses in the prevention and long-term management of various neuroinflammatory conditions.

### 4.1. PEA’s Neuroprotective Effects

PEA’s neuroprotective effects are due to its ability to modify MC, microglia and astrocyte activation [18,57]. It enhances microglial migration without promoting activation [18], thereby increasing resistance to infection without measurable pro-inflammatory effects [14]. This highlights PEA’s role as an immunomodulator rather than an immunosuppressant [14].

Other neuroprotective actions include inhibiting apoptosis and autophagy [16,18] by modulating the (bcl-2-associated X protein) bax/bcl-2 [56,58] and Akt/mTOR/p70S6K pathways [27,57]; limiting necrotic processes [18]; targeting NMDA receptors, thereby protecting cells against glutamate toxicity [28]; modulating synaptic homeostasis [28]; promoting neurogenesis [28]; and down-regulating the development of cerebral edema [18,20] and local inflammatory cascades [18]. PEA’s inhibition of pro-inflammatory cytokines likely contributes to its ability to prevent cortical spreading depression in pre-clinical models [111] and its therapeutic effects in migraine [60].

The above mechanisms correlate with PEA’s protective effects in ex vivo and murine models of Alzheimer’s Disease [14,15,16,17,19,54,55], Parkinson’s Disease [33,109], ASD [26,59], stroke [56] and traumatic brain injury [57,58,81]. In these models, chronic PEA administration (10–100 mg/kg) attenuated gliosis and neuroinflammation [15,16,17,19,33,54,55,56,57,58], protected against neuronal degeneration and apoptosis [15,16,17,19,33,54,55,56,57,58,81,109], rescued glutamate toxicity [16], inhibited oxidative stress processes [16,56,57] and improved behavioral [19,33,56,58,81], motor [33,57,109] and cognitive deficits [16,19,33,57].

As the endocannabinoid system regulates axonal growth, guidance, neurogenesis and behavior [59], these neurotrophic actions likely contribute to PEA’s benefits in maintaining cognitive, behavioral and mental health in individuals.

### 4.2. PEA’s Effects on Mood, Cognition and Behavior

ECs and NAEs are involved in the regulation of behavior, mood and cognition [32], and their levels are generally dysregulated in anxiety [30,32], depression [28,112,113], post-traumatic stress disorder (PTSD) [30] and ASD [26]. PEA levels are increased by acute psychosocial stress [30,112], presumably as a protective response to aversive and/or hazardous situations [30] and a modifier of the induced Cell Danger Response (CDR) [114]; but are unaffected [112] or reduced [28] in depression.

The therapeutic use of endocannabinoids in these disorders has produced promising results [26,28,30]. Analogously, many patients with PTSD self-medicate with cannabis [30] and antidepressants have been reported to increase brain PEA levels [28].

In one study, PEA levels were found to be higher in individuals suffering from PTSD compared to trauma-exposed individuals without PTSD and correlated with a greater symptom severity [30]. This suggests that endogenous NAE levels are insufficient to restore homeostasis in chronic aversive states, which would explain why PTSD patients self-medicate with cannabis [30] and why, in animal models of AD [33] and TBI [57,58,81], PEA administration improves memory function and reduces anxiety, aggressiveness and depression [57,58,59,81].

Interestingly PEA was shown to increase hippocampal neurogenesis and neuroplasticity in an established murine model of autism [33], and prevented the decrease in brain-derived neurotrophic factor (BDNF) and glial cell line-derived neurotrophic factor (GDNF) in a murine model of cerebral ischemia [56]. It also promotes the maturation of oligodendrocyte precursor cells [115]. Such nootropic effects may play a part in PEA’s cognitive-enhancing capacities. This further strengthens the potential use of PEA as a brain health-enhancing compound.

Much of the preclinical work has been replicated in human studies of variable quality. When used as an add-on therapy to levodopa in PD patients, PEA was found to improve motor and non-motor symptoms, including mood deficits, fatigue, sleep and mental tasks, after continuous administration for one year [32]. Caltagirone et al. investigated the effect of a co-ultramicronized composite containing 700 mg PEA and 70 mg luteolin in stroke patients, in combination with standard rehabilitation therapy, and found that PEA therapy for 60 days significantly improved neurological status, cognitive abilities and degree of spasticity, pain and independence [56]. De Palma et al. studied the effects of PEA treatment as an add-on to neuro-rehabilitation in a 12-year-old boy with traumatic brain injury in an intensive care unit [20]. At the end of the treatment period, the patient showed considerable improvements in motor and cognitive functions while being in a minimally conscious state. The authors deemed this improvement greater than had been seen in patients with similar conditions [20]. A double-blind, randomized, placebo-controlled study found that patients with major depressive disorder (MDD) receiving 600 mg PEA in addition to citalopram for 6 weeks twice daily demonstrated significantly greater improvements in depressive scores and symptoms compared to citalopram plus placebo group [28]. Three case studies on autistic children reported PEA administration (600–700 mg per day for several months to a year) improved behavior, sociability and cognition [26,59].

PEA’s ability to enhance neurogenesis [56] and facilitate synaptic plasticity [28,56] likely plays a role in improving behavior and cognition, but as depression and chronic pain can interfere with cognition, memory and decision making, PEA’s antidepressant, anxiolytic and analgesic actions are potentially also involved here [28,29,81].

PEA’s ability to target neuro-inflammation, pain, depression, anxiety and at the same time support neurogenesis and synaptic pruning makes it a viable therapeutic aid for brain disorders. The clinical data look promising, but further clinical trials are needed to confirm these findings. As all human studies to date report PEA to be well tolerated and without adverse effects [20,26,28,33,56], the potential role of PEA in enhancing brain health in clinically healthy populations becomes another key area of interest.

### 4.3. PEA’s Modulation of the Gut–Brain Axis

PEA’s ability to modulate gut health may have an impact on the CNS indirectly [84]. The gut–brain axis (GBA) describes the bidirectional communication between the central and enteric nervous system, linking cognitive and emotional centers of the brain with intestinal function [116]. Maintaining microbial symbiosis and gut barrier integrity is considered crucial for adequate brain development and neurological functioning [117]. “Leaky gut” caused by chronic intestinal inflammation [118], is likely associated with neuropsychiatric and neurodegenerative disorders via intestinal cytokine formation [84,103]. Additionally, bacterial toxins such as lipopolysaccharides (LPS) entering the systemic circulation through a compromised gut epithelium can traverse the blood–brain barrier (BBB) and act as ligands for receptors in the brain. These vectors initiate neuroinflammatory cascades, a hallmark of neurodegenerative disease and psychiatric disorders [85,119].

NAEs such as PEA are known to modulate peripheral and central processes of the GBA [84]. PEA’s anti-inflammatory action via PPAR-*α* in the gut epithelium has the potential to prevent neuroinflammatory responses by maintaining integrity of the gut barrier [22]. In a murine model of colitis, PEA attenuated inflammation and intestinal permeability and stimulated colonic cell proliferation in a PPAR-*α*- and CB2-dependent manner [22]. Other murine models of IBD found that higher levels of PEA levels were associated with less severe colonic inflammation and reduced proinflammatory cytokine production and immune infiltration [82].

PEA binds to GPR119 receptors in the gut and influences the secretion of satietogenic hormone GLP-1, which alleviates cognitive deficits in patients with a mood disorder [120]. Here is yet another way in which PEA may be supporting brain health and functionality.

## 5. PEA and Pain

Inflammation is a prominent characteristic of pain [121]. While inflammation and neuroinflammation are initially protective mechanisms, chronic inflammation creates an array of detrimental effects [122]. A key facet of neuroinflammation is the involvement of nonneuronal cells, including MCs, microglia and astrocytes. MCs and microglia are also said to cross-talk and activate each other to maintain and amplify the inflammatory condition [61,123,124]. These non-neuronal cells play a key role in central and peripheral sensitization, whereby previously neutral stimuli now elicit a painful response.

A large proportion of existing research into PEA is devoted to its analgesic properties. PEA has been investigated in pre-clinical models of inflammatory [62] and neuropathic pain [63,64], and its efficacy has been demonstrated in clinical conditions including osteoarthritis and joint pain [29,65], neuropathic pain [66,67,68], post-operative pain [69], fibromyalgia [70] and endometriosis [125]

PEA is endogenously produced on-demand in all tissues, as a protective response to injury, inflammation and pain. [27,37,124]. When pain is protracted, however, PEA ‘exhaustion’ may develop. Chronic inflammatory conditions create lower levels of PEA [37,124]. The exogenous administration of PEA may in such cases serve to replenish levels of endogenous PEA, restoring its protective, anti-inflammatory and analgesic effects. Interestingly, a recent report presented the case of an individual with hypoalgesia resulting from an inability to degrade PEA and the analogous fatty acid amides [126].

PEA’s analgesic effects operate via multiple pathways. PEA acts directly on PPAR-*α* and GPR55 receptors and indirectly on CB1, CB2 and TRPV1 receptors [104,127,128,129,130]. It suppresses inflammation by inhibiting MC activation, down-regulates mediators such as NGF, COX-2, TNF-α and iNOS [25,41,131] and inhibits microglia and astrocyte activation [64,132,133]. In chronic inflammatory states, this allows PEA to preserve peripheral nerve morphology and reduce endoneurial edema and macrophage infiltrates [64,134].

Currently available analgesics carry risks of gastrointestinal, hepatic, cardiovascular and renal damage, which increase when managing long-standing conditions [135]. PEA’s analgesic efficacy and safety make it a promising alternative candidate in the management of chronic pain and pain in vulnerable individuals.

### 5.1. PEA’s Effects on Primary Headache

The lifelong prevalence of headache is 96% [136]. The bulk of these are primary headache disorders including migraine and tension-type headaches (TTH), which can occur in otherwise healthy individuals as isolated episodic incidences or chronically [137]. These constitute a significant health burden worldwide [138].

Central sensitization and inadequate endogenous pain control are thought to be involved in chronic TTH. The current understanding implicates nociception from pericranial myofascial tissues [139]. Early stages of migraine are caused by trigeminal nociceptor activation, as a result of neurovascular inflammation in the meninges and around cranial blood vessels [140]. Sensitization of the perivascular trigeminal nerve terminals then elicit pain responses to previously non-painful stimuli [140]. Meningeal nociceptors are believed to be activated locally by resident MCs of the dura mater and associated glial cells, which release pronociceptive and proinflammatory mediators [71]. As PEA down-regulates this process, it presents a novel approach for primary headache treatment.

PEA’s efficacy in treating migraine has been investigated in several studies. Dalla Volta and colleagues reported that the sublingual administration of 600 mg um-PEA b.i.d. for 3 months caused a reduction in headache frequency, duration and intensity and number of analgesics taken per month in patients with migraine without aura [72]. A second study showed that in patients suffering from migraine with aura treated with non-steroidal anti-inflammatory drugs (NSAIDs), the add-on chronic administration of 1200 mg/day um-PEA caused a statistically significant and time-dependent reduction in pain [60]. It reduced the number of attacks per month and days of pain during each attack. A similar finding was demonstrated in an open-label study of a pediatric population with migraines without aura. Daily PEA supplementation of 600 mg for 3 months caused a reduction in headache attack frequency, attack intensity and percentage of patients with severe attacks [141].

These findings indicate a potential use of PEA as a migraine prophylactic and a possible treatment for TTH.

### 5.2. PEA’s Effects on Menstrual Pain

Dysmenorrhea is the most prevalent gynecologic condition in women of reproductive age and a leading cause of chronic pelvic pain [142]. Primary dysmenorrhea has no clear underlying pathology and affects otherwise healthy individuals. It is characterized by painful cramping in the pelvic and lower abdominal region, which may radiate to the lower back and legs, and can also cause nausea, vomiting, diarrhea and headache. It is associated with anxiety and depression and contributes greatly to incidence of school or work absenteeism [143,144]. Despite high prevalence and levels of discomfort, medical treatment is not commonly sought, for predominantly cultural reasons.

The decrease in progesterone immediately prior to menstruation leads to a release of fatty acids including arachidonic acid from uterine cells and the production of mediators such as prostaglandin F_2α_ (PGF_2α_) and prostaglandin E_2_ (PGE_2_), which lead to myometrial contraction and vasoconstriction, causing local ischemia and pain [145]. The menstrual fluid of women with dysmenorrhea has higher levels of these prostaglandins than that of eumenorrheic women [73], with a direct correlation between severity of dysmenorrheic symptoms and prostaglandin levels. These are highest during the first two days of menstruation, which coincides with the period of greatest pain. MC degranulation may play a role in peripheral nerve sensitization and pelvic pain [74], which, if recurrent, may cause central sensitization and increase susceptibility to other chronic pain conditions. As PEA down-regulates COX-2 and prevents the recruitment and activation of MCs, it would be expected to mitigate the hyperinflammatory state present in dysmenorrhea. In one trial, the combined administration of PEA-transpolydatin (400 mg + 40 mg) for 10 days to young women with primary dysmenorrhea significantly reduced pelvic pain compared to placebo [74].

Secondary dysmenorrhea can be caused by endometriosis, which is increasingly viewed as a chronic inflammatory disorder due to the involvement of MC degranulation in proximity to nerves in lesion sites [125]. In a murine model of endometriosis plus ureteral calculosis, administration of 10 mg/kg/d PEA significantly reduced viscero-visceral hyperalgesia, likely through the down-modulation of MC activity in endometrial cysts, thereby reducing central sensitization [75]. Clinical trials have already demonstrated the beneficial effect of PEA plus polydatin or transpolydatin, in the treatment of secondary dysmenorrhea associated with endometriosis [76,77,125,146].

Though NSAIDs are commonly used in the management of primary headache pain and primary dysmenorrhea, their adverse effect profiles are a concern and their chronic use may cause paradoxical overuse headache. PEA is devoid of safety concerns and offers a more physiological alternative, especially for chronic and/or recurrent pain related to these two conditions. It may also be safely used as an add-on, as there are no known interactions, and prophylactically, which would be expected to reduce the risk of central sensitization.

## 6. PEA and Joint Health

Musculoskeletal pain makes a significant contribution to the global burden of disease [147]. Osteoarthritis (OA) is the leading form of joint pain and disability worldwide and may cause acute, recurring or chronic pain [148]. Although more prevalent in older adults, younger individuals are also susceptible [149,150].

In OA, joint pathology is primarily attributed to the progressive degradation of articular cartilage and a local imbalance of pro-inflammatory and pro-resolving mediators [151,152]. The local release of pro-inflammatory cytokines, growth factors, and neurotransmitters increases nociception [148], and OA patients often report symptoms of neuropathic pain, such as burning, tingling and numbness [153].

In the affected joints, chondrocytes become unable to sustain a balance between synthesis and degradation of the extracellular matrix, leading to progressive cartilage thinning [154]. Other features include the thickening of subchondral bone, osteophyte formation, ligament degeneration and synovial inflammation [155]. Because the degree of pain does not always correlate with the severity of joint damage or inflammation [153,156], there is a strong case for relinquishing simple NSAIDs for more holistic therapeutic strategies.

### 6.1. Current Solutions for Joint Health

Currently, paracetamol, acetaminophen and NSAIDs are the most common treatments used to manage joint pain [157]. These drugs are associated with adverse effects including gastrointestinal bleeding, cardiovascular side effects and gut dysbiosis [158,159], and are in any case not particularly effective [160].

Glucosamine, chondroitin and MSM are used in joint health formulations and may provide some structural benefits to bone and joints [161,162], but suffer from high dose requirements, poor bioavailability, delayed onset of action and uncertain efficacy [65,78,163,164,165].

### 6.2. PEA’s Effects on Joint Health

PEA is a promising alternative compound for relieving joint pain. One clinical trial found that PEA was more effective than ibuprofen in relieving the pain of temporomandibular joint (TMJ) osteoarthritis [65]. A second placebo-controlled trial [29] demonstrated the effectiveness of PEA on patients with knee osteoarthritis. In this 8-week clinical study, researchers found a dose-dependent improvement in joint pain, stiffness and function, as measured by the Western Ontario and McMaster Universities Osteoarthritis Index (WOMAC), in those who consumed a high-bioavailability form of PE. By week 8, joint pain was reduced by 40% (300 mg) and by 49.5% (600 mg), with significantly reduced use of rescue medication.

In a third double-blind, randomized, placebo-controlled study [166], a daily dose of 350 mg high-bioavailability PEA (containing NLT 315 mg PEA) was assessed for efficacy in 80 individuals experiencing general joint pain. A visual analogue scale was used to self-assess joint pain in the morning and evening. The active group experienced a significant reduction in joint pain after 14 days compared to placebo. Joint pain was significantly reduced as early as 3 days.

All three studies are in line with existing literature on PEA’s substantial therapeutic index, which makes it an interesting candidate for the first line treatment of joint pain. Moreover, due to its positive effects on tissue damage resolution and healing [127,167,168,169,170,171,172], it offers the prospect of a more effective way of preserving joint structure and function.

## 7. PEA and Exercise Recovery

Musculo-skeletal health is important not only to those who engage in sporting and athletic activities but also to the much larger proportion of individuals who choose to remain physically active as part of a healthy lifestyle [173]. Injuries are a part of life, and a primary reason why people stop exercising. Rest and recuperation are often indicated but safe analgesic/anti-inflammatory support is also desirable, especially in the ageing population or occasional athletes [29].

Exercise-induced muscle injury can occur after a bout of unaccustomed exercise [174] and lead to unfavorable outcomes such as loss of force and longer periods of recovery [175]. If recovery protocols are optimized, individuals are able to return to training and competition more quickly and there is evidence that they can also train and subsequently perform at higher intensities [176].

Exercise-induced muscle damage (EIMD) is characterized by transient ultrastructural myofibrillar disruption, delayed onset muscle soreness (DOMS), swelling, reduced range of motion, activation of innate immune cells such as macrophages and mast cells [177] and markers associated with muscle damage such as myoglobin, creatine kinase and lactate dehydrogenase, or an amalgamation of these [178]. Although the activation of muscle stem cells leads to a physiological remodeling [177], in the short term, significant amounts of myoglobin and lactate leaking into the circulation disrupt muscle cells [179] and impair exercise performance [180].

Nutritional strategies that could reduce EIMD and accelerate recovery without impeding remodeling would be highly desirable.

### 7.1. Current Solutions for Exercise Recovery

NSAIDS are commonly used to relieve symptoms of exercise-induced muscle damage, particularly by athletes who use four times more painkillers than the general population [181]. However, the long-term use of NSAIDS impairs muscle adaptation to exercise and carries the risk of potentially serious adverse side effects [182].

Cannabidiol (CBD) has recently gained interest as a potential tool to accelerate recovery. Reports are mixed, and issues of legality, toxicity, doping risks and the limited research supporting its application in healthy populations have hindered the adoption of CBD as a recovery strategy [183,184,185,186,187].

### 7.2. PEA’s Effect on Muscle/Exercise Recovery

PEA’s application to an exercising population and potential for muscle recovery is not well understood. To date, only one clinical trial has assessed the impact of PEA on recovery from muscle-damaging exercise [31]. This study found that the group consuming 176.5 mg of a high-bioavailability form of PEA (containing NLT 158 mg PEA) in liquid form had significantly lower myoglobin and blood lactate levels than the placebo group. These shifts signify reduced muscle damage and increased aerobic energy metabolism, respectively, findings associated with enhanced recovery and the ability to maintain higher exercise intensities for longer.

The results are in line with PEA’s ability to reduce muscle protein breakdown as shown by a significant increase in protein kinase B (Akt) phosphorylation, a kinase known to induce protein synthesis [188], and the ability to clear myoglobin from the circulation [31]. The fall in lactate levels is not yet fully understood but reflects decreased lactate production and/or increased uptake into surrounding tissues. The overall results indicate that PEA may enhance concurrent training adaptations after exercise, and are being further investigated. It may be particularly useful for individuals who require rapid recovery between successive acute bouts of exercise (e.g., competition, two-a-day workouts) [189].

## 8. PEA and Sleep

Circadian rhythms underpin survival, bodily homeostasis and motor and cognitive functions [190,191,192]. These include the internal regulation of the sleep–wake cycle and the quality of sleep. Reduction in sleep quality elicits short- and long-term adverse consequences, including longer periods of elevated sympathetic nervous system activity and waking to physical and psychological stressors [34,193]. Protracted changes in the sleep–wake rhythm may cause synaptic changes leading to neurobehavioral deficits, lapses in attention, slowed working memory, verbal fluency and cognitive ability and mood disorder such as depression, and even influence all-cause mortality [194,195,196].

Conversely, environmental and psychosocial stressors as well as various medical conditions can give rise to a range of sleep disorders [197,198]. The deleterious impact of chronic pain on sleep quality has been extensively documented. Chronic pain, such as that caused by neuropathic pain conditions, joint pain, peripheral diabetic neuropathy and carpal tunnel syndrome (CTS) are all associated with poor sleep quality [34,198,199].

Prescription sedatives and tranquilizers are widely used for treating sleep disorders [200], but carry risks of addiction and adverse effects. The cannabinoids may be useful when used short-term, but withdrawal after chronic cannabinoid use has been shown to cause sleep deprivation in animal models and in humans [190]. There is a need for natural products that improve sleep quality without the adverse effects listed above.

PEA’s Modulation of Sleep

Chronic pain is linked to impaired sleep. Environmental stress, depression and anxiety states alter pain perception [201] and are linked to high sleep reactivity which, similar to pain, can also lead to insomnia [197]. A decline in AEA is thought to amplify stress responses and activate the hypothalamic-pituitary-adrenal (HPA) axis, leading to an increase in anxiety behavior [202].

Given the common linkage between pain, stress and impaired sleep, PEA’s analgesic effects [29,34,79] and anxiolytic properties as demonstrated in pre-clinical models [80,203,204], exerted at least in part via TRPV1 channel and AEA regulation [203], make it a logical candidate for use in pain- and stress-associated sleep disorders. Its entourage effect on AEA may also attenuate sleep disorders caused by stress and anxiety.

There is some recent clinical evidence of anxiolysis. An acute dose of PEA (1.2 g/day) used together with citalopram was shown to improve the symptoms of severe depression [28]. A recent double-blind randomized placebo-controlled study assessing enhanced bioavailability PEA in osteoarthritis patients found that it reduced stress and anxiety in patients along with knee pain [29]. In one study of carpal tunnel syndrome, which is associated with emotional distress [205], 1200 mg of ultra-micronized PEA during pre- and post-surgery periods significantly improved the sleep–wake rhythm and overall quality of sleep [34].

High levels of AEA are linked to wakefulness in healthy individuals and declining levels in the elderly are associated with circadian rhythm imbalance and cognitive impairment [190]. Via the entourage effect, PEA may, therefore, support the sleep–wake balance in healthy adults [206].

While further research is required, PEA appears to have potential as a supplement suited to long-term use for holistic improvement in the quality of sleep and life for many individuals. PEA’s combined analgesic, anxiolytic and antidepressant effects differentiate it from any other sleep aid currently in use, and make it an attractive alternative to current treatments.

## 9. Overcoming Challenges in PEA’s Bioavailability

PEA is lipophilic in nature and almost insoluble in water [9], and its poor solubility and bioavailability has limited the development of nutraceutical applications.

Rate-limiting factors for absorption include dissolution rate and the aqueous barrier of the gastrointestinal lumen, and are influenced by PEA’s lipophilicity and particle size [62]. Once absorbed, PEA is rapidly metabolized and excreted [23]. It has a relatively short half-life; levels of PEA in human plasma return to baseline within two hours of ingestion [104].

Due to these considerations, PEA in supplements is usually emulsified [27], micronized [62] or utilized as a specialized delivery system [207] to improve bioavailability and functionality.

Mechanical comminution (i.e., crushing, grinding and milling) is commonly used to reduce the size of large crystals in order to enhance dissolution and increase bioavailability [208]. Micronized and ultra-micronized forms of PEA have been used in animals [209] and humans [65,104], but comparative pharmacokinetic data showing any improvement with these micronized forms are non-existent.

The novel crystalline dispersion technology (LipiSperse^®^ [Pharmako Biotechnologies Pty Ltd., Frenchs Forest, Australia]) uses a combination of surfactants, polar lipids and solvents to increase the wettability of lipophilic substances in aqueous environments [210]. By embedding amphiphiles into the surface of lipophilic molecules, LipiSperse^®^ decreases the contact angle with water, reduces the surface tension between particles and acts as a dispersing agent. The prevention of agglomeration increases the specific surface area of the lipophilic substance in the GIT, thus enhancing absorption.

This technology has previously been shown to improve the bioavailability of trans-resveratrol [211] and curcuminoids [210] in single equivalent dose, randomized and double-blinded studies, with optional cross-over [210]. 

To our knowledge, only one study has compared the absorption of different forms of PEA. This study compared the bioavailability of 300 mg of PEA using LipiSperse^®^ delivery technology against 300 mg of non-micronized PEA, and used a parallel, double-blinded trial design. While the AUC was significant in both groups, LipiSperse^®^-enhanced PEA achieved plasma PEA levels 1.75 times higher than those achieved with non-micronized PEA [207]. Both formulations showed twin plasma peaks at 90 and 180 min for LipiSperse^®^-enhanced PEA; and 70 and 120 min for non-micronized PEA. This pattern has been recorded elsewhere, and indicates enterohepatic recycling. The Lipisperse^®^-PEA treated group showed earlier first peaks (105 min vs. 125 min) and this, together with the superior AUC data, suggests that the Lipisperse^®^ delivery technology will provide faster onset and a more sustained effect than other forms of PEA.

The increased absorption and bioavailability provided by LipiSperse^®^ leads to higher active concentration of PEA, enabling lower dosages in nutraceutical formulations compared to non-micronized PEA [207]. The cold-water dispersibility also allows PEA to be incorporated into a broader range of delivery formats (i.e., effervescent tablets, powders and gels).

Understanding of exogenous PEA pharmacokinetics is still at an early stage [212]. Future research should assess the precise tissue distribution and site of metabolism of PEA in order to establish true pharmacokinetic profiles of non-micronized, micronized and ultra-micronized PEA using dispersion technology.

## 10. Concluding Remarks

In the aftermath of the nutrition transition, public health has markedly declined. Dietary improvement, stress reduction, exercise and improved socialization are all widely recommended; however, there is clearly also a role for judicious supplementation.

This review documented the primarily protective effects of endogenous PEA and the diverse benefits of exogenous PEA in a range of chronic disorders and minor ailments, and its excellent safety record. PEA offers improved quality of life in many instances, and appears to be partially gero-suppressant. Ongoing and pending clinical trials investigating the health benefits of PEA in healthy adult populations will provide further answers.

## Figures and Tables

**Table 1 ijms-22-05305-t001:** Overview of PEA as a potential lifestyle management solution.

Effects	Indications	References
Anti-inflammatory/prophylactic	Anti-ageing, immuno-enhancement	[38,39,40,41,42,43,44]
Anti-inflammatory	Allergic reactions, Exercise recovery, Brain health	[24,25,31,45,46,47,48,49,50,51,52,53,54,55,56,57,58,59,60]
Antinociceptive/Anti-inflammatory	Pain, Joint health, Sleep, Brain health	[15,16,17,19,26,27,29,34,54,55,56,57,58,59,60,61,62,63,64,65,66,67,68,69,70,71,72,73,74,75,76,77,78]
Anti-inflammatory/anxiolytic	Brain health, Sleep	[27,29,34,35,79,80]
Nootropic	Brain health	[16,19,26,27,31,33,56,57,58,59,81,82]
Gut microbiome modulation	Immunity, Brain health, Various	[82,83,84,85]

**Table 2 ijms-22-05305-t002:** Summarization of PEA’s immunomodulatory effects.

Effects	Mechanisms of Action	References
Anti-inflammatory/prophylactic	Regulation of macrophages and cytokines via PPAR-*α*	[38,39]
Anti-inflammatory	Control of MC degranulation	[40,41]
Prophylactic	Nonspecific bacterial resistance	[42,43,44]
Entourage	Modulation of migration and phagocytic activities of macrophages and microglial cells	[91,92,93]
Anti-inflammatory/Gut-derived immunity modulation	Maintenance of gut barrier and microbiome integrity via PPAR-*α*	[83,84]

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
