# Peer review of "Palmitoylethanolamide: A Natural Compound for Health Management"

_ijms, 2021, doi:10.3390/ijms22105305_

Round 1

Reviewer 1 Report

This rather well written and rather comprehensive review highlights many of the potential therapeutic benefits of the endocannabinoid N-acylethanolamine palmitoylethanolamide (PEA).  

Minor issues

  • the title has "Lifestyle Management" in the title, but the article deals with few issues that can be ascribed to lifestyle.  Indeed, while PEA has some evidence supporting it utility to combat obesity (a clinical complication strongly associated to wester lifestyles) and associated issues (especially obesity-associated inflammation) this topic is not discussed.  While it is not crucial that such topics be added to the review, without them, the term "lifestyle" in the title seems strange
  • a dedicated section on PEA pharmacokinetics/ADME could benefit the review given the section on bioavailability
  • calling PEA a prebiotic is using a very broad definition of the the term prebiotic.  It has not been shown (to the best of this reviewers knowledge) to  be used as an energy source for bacteria that would result in altered abundance of various taxa.  Many pharmaceuticals have effects on the microbiome, though they are not considered prebiotics
  • Line 355-457, A. muc. does not bind to GPR119 (that we know of), the authors may have meant PEA binding to GPR119
  • Levagen is not a probiotic, which refers to beneficial bacteria
  • line 215, which biosynthesizing enzyme?
  • the effective doses of PEA are rarely mentioned(though conspicuously are often mentioned in Levagen studies), even when dose-dependent results are reported.  Their inclusion where appropriate would be beneficial to understand the differential activity within different conditions
  • the authors should go over the reference list.  For example ref 33, which is a clinical PD study is referred to several times as a preclinical study, an autism study and an Alzheimer's study.
  • line 378-379, the FAAH mutation referred to here affects AEA, OEA and not just PEA levels and AEA is believed to be a major contributor to the reported hypalgesia
  • section 6.3 seems lacking and out of place
  • line 644-645, the concept of the mentioned formulation "requiring lower doses" should be clarified.  Do the authors mean for obtaining the same circulating level of PEA within the blood (which is easily justified) or for obtaining the same beneficial/clinical effects (which would be speculation without any evidence to support this).

Major issue

The major concern of this reviewer that would preclude the article's publication in it's current form the the apparent bias towards Levagen.  While the authors acknowledge that several formulations of PEA are available on the market, they are only mentioned in passing and rarely in conjunction with specific pre-clinical or clinical studies, while Levagen is mentioned a total of 33 times within the review.  While the authors have stated their conflict of interest, the article must be revised so that all commercial formulations are treated fairly with respect to their being mentioned in conjunction with specific results.  Related to this issue, in lines 177–180, Gencor group's PEA formulation resulting in "promising results" needs substantiation or a citation if the group successfully published the preliminary findings or the trial design.

Author Response

Almost all the changes requested have been made. In order of the list of requested changes, this includes a changed title, an acknowledgement of the scarcity of pharmacokinetic data, the deletion of any prebiotic activity and rectification of the error made re. binding to GPR119 (the reviewer was absolutely right, this was an error on our part).

The biosynthesising enzyme formerly on l 215 has been specified, the doses of PEA used in various trials have been cited, all references have been checked and amended where appropriate.

At line 378-379, the FAAH mutation has been re-written to include the FAA analogues; and section 6.3 has been amalgamated with the larger section on exercise, muscle pain and recovery.

At former lines 644-645, the statement made re dosages has been refined, and we have attempted to establish parity or near-parity between PEA brands.

We look forward to your comments, and thank you for your patience.

Reviewer 2 Report

Manuscript ID: ijms-1147573

Type of manuscript: Review

          The review article titled “Palmitoylethanolamide: A Natural Solution for Lifestyle Management” by Paul Clayton et al have demonstrated the use of palmitoylethanolamide as functional food for therapy of lifestyle diseases. The review discusses the effects of exogenous PEA an endocannabinoid-like lipid mediator with extensively documented anti-inflammatory, analgesic, antimicrobial, immunomodulatory and neuroprotective effects. The various studies related to boosting immunity against viral and bacterial infections, modulation of inflammatory mediators, pain relief mechanism, neuroprotection and overall brain health, exercise recovery and joint health and sleep modulation are described extensively. 

          The review has critically evaluated the existing work, and provided the material more easily available to those not expert in the area through clear text. The PEA’s poor oral bioavailability has been tackled. Considering that PEA has several benefits as discussed, this section needs more discussion. Although the pharmacokinetic data for such studies is not available, the current review can stimulate further research in this area.

General comments-

In the references section, include the missing DOI for the references.

Author Response

We thank you for your comments, and have submitted a new version of the paper which includes additions made in response to both your feedback and that of your fellow reviewer. All track changes are visible, as per request of the journal. DOI data has been included in all cases where this was available.

Sincerely,

Paul Clayton

Round 2

Reviewer 1 Report

Many of the concerns of this reviewer have been addressed, however, the entire paper from section 7 onwards has been highlighted making it very difficult to detect changes in the manuscript.  Further, there are still issues with the references that must be corrected.  While this reviewer has not gone through them in detail I note that #112 and #165 are no longer mentioned in the text.  It is curious that both of these studies relate to the Normast formulation of PEA. Which brings me to my only major concern precluding the publication of this manuscript; while the authors have halved the number of times they mention Levagen, they still fail to consistently mention other branded PEA formulations.  For example,

"There is also an intriguing case report of a 13-year-old child with autism, 233
presenting with significant atopic illness including chronic eczema, allergic rhinitis and 234
asthma. The physicians reported that daily oral administration of PEA for one month (first 235
at 300 mg/day and later increased to 600 mg/day) resulted in a marked reduction in allergy 236
stigmata, skin eczema and urticaria [26]."

This PEA used here was, coincidentally, Normast, then two sentences later the authors write.

"Two studies are underway investigating the antiallergic efficacy 238
of Levagen®+ in allergic rhinitis and eczema."

This appears to be a blatant bias, notwithstanding the clear statement of conflict of interest.  All commercial formulations must be treated equally within the review; either mention the commercial names when referring to the studies in which they were used or not.  There can not be a skewing in favour of one formulation over another.

Author Response

With the exception of the section on comparative pharmacokinetics where it seemed necessary to identify the technology and the products being compared, all brand names have been removed. 

All references have been checked and a small number of corrections made regarding numbering.

Some minor grammatical and spelling errors have been rectified.

We hope the document is now compliant!